# Incidence of Diabetic Retinopathy in Individuals with Type 2 Diabetes: A Study Using Real-World Data

**DOI:** 10.3390/jcm13237083

**Published:** 2024-11-23

**Authors:** Carlos Hernández-Teixidó, Joan Barrot de la Puente, Sònia Miravet Jiménez, Berta Fernández-Camins, Didac Mauricio, Pedro Romero Aroca, Bogdan Vlacho, Josep Franch-Nadal

**Affiliations:** 1Primary Health Care Centre Burguillos del Cerro, Servicio Extremeño de Salud, 06370 Badajoz, Spain; carlos.teixido92@gmail.com; 2RedGDPS Foundation, 08204 Sabadell, Spain; joanbarrot@hotmail.com (J.B.d.l.P.); smiravet12@gmail.com (S.M.J.);; 3Departament of Medicine, Universitat de Barcelona, 08036 Barcelona, Spain; 4DAP-Cat Group, Unitat de Suport a la Recerca Barcelona, Fundació Institut Universitari per a la Recerca a l’Atenció Primària de Salut Jordi Gol i Gurina (IDIAPJGol), 08007 Barcelona, Spain; 5Primary Health Care Center Dr. Jordi Nadal i Fàbregas (Salt), Gerència d’Atenció Primària, Institut Català de la Salut, 17007 Girona, Spain; 6Primary Health Care Center Martorell, Gerència d’Atenció Primària Baix Llobregat, Institut Català de la Salut, 08007 Barcelona, Spain; 7Institut de Recerca Hospital de la Santa Creu i Sant Pau, 08041 Barcelona, Spain; 8Department of Endocrinology and Nutrition, Hospital de la Santa Creu i Sant Pau, 08025 Barcelona, Spain; didacmauricio@gmail.com; 9CIBER of Diabetes and Associated Metabolic Diseases (CIBERDEM, ID CB15/00071), Instituto de Salud Carlos III (ISCIII), 28029 Madrid, Spain; 10Department of Medicine, University of Vic-Central University of Catalonia, 08500 Vic, Spain; 11Ophthalmology Service, University Hospital Sant Joan, 43202 Reus, Spain; romeropere@gmail.com; 12Institut de Investigacio Sanitaria Pere Virgili (IISPV), University of Rovira and Virgili, 43002 Tarragona, Spain; 13Primary Health Care Center Raval Sud, Gerència d’Àmbit d’Atenció Primària Barcelona Ciutat, Institut Català de la Salut, 08007 Barcelona, Spain

**Keywords:** diabetic retinopathy, incidence, type 2 diabetes mellitus, microvascular complications, primary care

## Abstract

**Background/Objectives**: This study aimed to assess the incidence of diabetic retinopathy (DR) in patients with type 2 diabetes (T2DM) treated in primary-care settings in Catalonia, Spain, and identify key risk factors associated with DR development. **Methods**: A retrospective cohort study was conducted using the SIDIAP (System for Research and Development in Primary Care) database. Patients aged 30–90 with T2DM who underwent retinal screening between 2010 and 2015 were included. Multivariable Cox regression analysis was used to assess the impact of clinical variables, including HbA1c levels, diabetes duration, and comorbidities, on DR incidence. **Results**: This study included 146,506 patients, with a mean follow-up time of 6.96 years. During this period, 4.7% of the patients developed DR, resulting in an incidence rate of 6.99 per 1000 person-years. Higher HbA1c levels were strongly associated with an increased DR risk, with patients with HbA1c > 10% having more than four times the risk compared to those with HbA1c levels < 7% (hazard ratio: 4.23; 95% CI: 3.90–4.58). Other significant risk factors for DR included greater diabetes duration, male sex, ex-smoker status, macrovascular disease, and chronic kidney disease. In contrast, obesity appeared to be a protective factor against DR, with an HR of 0.93 (95% CI: 0.89–0.98). **Conclusions**: In our real-world setting, the incidence rate of DR was 6.99 per 1000 person-years. Poor glycemic control, especially HbA1c > 10%, and prolonged diabetes duration were key risk factors. Effective management of these factors is crucial in preventing DR progression. Regular retinal screenings in primary care play a vital role in early detection and reducing the DR burden for T2DM patients.

## 1. Introduction

Diabetes mellitus (DM) is one of the fastest-growing chronic diseases in terms of global prevalence [1,2]. The International Diabetes Federation (IDF) estimates that 10.5% of adults between 20 and 79 years old have diabetes. The IDF also estimates that 643 million adults will live with diabetes in 2030 (11.3% of the population), and this number will reach 783 million (12.2%) by 2045. Thus, DM-related complications, including diabetic retinopathy (DR), are an increasingly serious health problem [3].

DR is associated with small-vessel disease and neurodegenerative complications [4]. It is the most frequent complication of DM and the leading cause of visual impairment and preventable blindness in the working-age population. Moreover, it is considered a serious public health concern that is reaching alarming rates throughout the world [5]. The prevalence of DR increases with disease duration and poorer glycemic and blood pressure control [6,7,8].

People with DR experience difficulties in activities of daily living and have lower quality of life [9]. In addition, DR has been associated with micro- and macrovascular complications of DM, macrovascular comorbidities (coronary heart disease and cerebrovascular accidents), and mortality from all causes [10,11,12,13,14,15,16,17,18]. This corresponds to a significant consumption of healthcare resources. Average healthcare costs increase considerably with the severity of DR, suggesting that preventing the progression of DR may reduce healthcare costs [19,20].

According to recently published studies, there is considerable variability in the prevalence of DR. The IDF reported that the global prevalence of any DR between 2015 and 2018 was 27.0%; the most common (25.2%) form was mild non-proliferative retinopathy (NPDR), followed by diabetic macular edema (DME) (4.6%) and proliferative diabetic retinopathy (PDR) (1.4%) [21]. The prevalence of any type of DR in Spain was 15.28% in 2020 (and 1.92% for sight-threatening diabetic retinopathy [STDR]—PDR and/or DME), while in Catalonia (Northeast Spain), the prevalence of DR was 12.3% [22,23]. A decrease in the prevalence of DR in developed countries has been reported, which may be due to earlier diagnoses and improved care for patients with T2DM resulting in a reduction in T2DM-related complications [24].

While data on the prevalence of DR have been reported in many studies, evidence on its incidence is scarcer. A systematic review of studies from 1980 to 2019 reported that the incidence of DR ranged from 2.2% to 12.7% across eight studies from different regions [25]. Other systematic reviews have reported a pooled annual incidence in Europe of any DR in T2DM equal to 4.6% (95% confidence interval [CI]: 2.3–8.8%) [26] and a pooled incidence of DR equal to 3.83% and 0.41% for STDR in Spain [23].

Updating the prevalence and incidence of DR is an essential requirement to support the planning and allocation of resources needed in the future to help alleviate the burden of preventable vision loss and blindness associated with diabetes. This study aimed to assess the incidence of DR and risk factors among people with T2DM in a primary healthcare setting in Catalonia (Northeast Spain).

## 2. Materials and Methods

### 2.1. Study Design, Setting and Population

We conducted a retrospective cohort study on people diagnosed with T2DM who received care at primary healthcare centers within the Catalonian Health Institute (ICS). The ICS provides public healthcare coverage to 83% of the population of Catalonia (Northeastern Spain). Data for our study were obtained from the Population Database of the System for Research and Development in Primary Care (SIDIAP https://www.sidiap.org/). This database contains routinely collected healthcare information such as demographic information, appointment dates with healthcare providers, clinical diagnoses, clinical variables, prescribed medications, referrals to specialists and hospitals, laboratory test results, and dispensed medications. This database is a well-validated data source for the study of diabetes in Spain [27,28].

The data were collected from 1 January 2010 to 31 December 2021, and the Index date was the date of the first retinography scan during the inclusion period between 2010 and 2015. The follow-up period was defined to be until the data extraction end date (31 December 2021) or the event of discontinuation (death or any other database dropout).

### 2.2. Definition of Eligibility Criteria

Our study included all patients between 30 and 90 years of age who were diagnosed with T2DM, according to International Classification of Diseases (ICD-10) diagnostic codes and sub-codes (E11 and E14), and had undergone fundus photography with a non-mydriatic camera to determine the presence of DR. Inclusion criteria required that all retinal photographs were classified as showing no apparent retinopathy (NDR) during the inclusion period. We excluded individuals with other types of diabetes (type 1 diabetes, gestational diabetes, secondary, or other) as well as those without retinal photography data or codification.

### 2.3. Definition of Variables

We assessed the initial absence or presence of DR using retinal photography. Two digital images were taken of each eye: one at 45° centered midway between the macula and the optic disc, and the other centered on the macula. The Early Treatment of Diabetic Retinopathy Study (ETDRS) classification has been considered the “gold standard”, but its clinical application is limited by its complexity [29,30].

The severity of DR was categorized according to the International Clinical Diabetic Retinopathy (ICDR) Severity Scale [31]. This new classification is simple to use, based on scientific evidence, and the most commonly used classification system worldwide.

The presence of any DR was defined as the presence of abnormal (pathologic) fundus photography results. The severity of DR was reported as follows: NDR, mild NPDR, moderate NPDR, severe NPDR (including severe and highly severe), PDR, and DME. These DR severity grades were considered events during the follow-up period. The eye most severely affected was used to assign each patient a DR category. Retinal photography was performed by trained personnel using a non-mydriatic camera, and the results were included in the electronic medical record.

We also collected data on sociodemographic characteristics (age and sex), T2DM-related factors (duration of diabetes and glycated hemoglobin levels [A1c]), clinical variables (systolic and diastolic blood pressure, body mass index [BMI], and smoking status), laboratory parameters (lipid profile and renal profile), cardiovascular risk factors (obesity, hypertension, dyslipidemia, and smoking), and concomitant medication (antihypertensive, antiplatelet, lipid-lowering, and antidiabetic drugs). The antidiabetic treatment was classified using the following categories: no drugs, non-insulin antidiabetic drugs (NIAD), only insulin, or combined NIAD+ insulin.

Renal profile tests included serum creatinine levels and urinary albumin-to-creatinine ratio (UACR). Chronic kidney disease (CKD) was defined as UACR > 30 mg/g and/or eGFR < 60 mL/min/1.73 m^2^.

### 2.4. Statistical Analysis

We used descriptive statistics to analyze the variables. We calculated the absolute and relative frequencies of the qualitative variables and their means and standard deviations, or medians and interquartile ranges, according to the variables’ characteristics.

At follow-ups, for the primary study event, we calculated the number of subjects and events for each stage of DR, the time till the event occurred (years), cumulative incidence, and incidence rates (person/year). To assess the probability of occurrence, we used proportional hazards regression analysis. The hazard ratios (HRs) for the primary outcome event were calculated with corresponding 95% confidence intervals (CIs), and statistical significance was established as *p* < 0.05. In the final multivariate model, adjustments were made for the following variables: age (years), sex, diabetes duration (years), HbA1c, tobacco use, body mass index (BMI), hypertension, dyslipidemia, and macrovascular disease. Specifically, age, diabetes duration, HbA1c, and BMI were included as continuous parameters, while sex, tobacco use, hypertension, dyslipidemia, and macrovascular disease were treated as categorical parameters. This approach ensured that both continuous and categorical variables were properly accounted for in the analysis. We also performed a sensitivity analysis with the estimates from different models and stratification for diabetes duration and HbA1c.

We used Kaplan–Meier to graphically visualize the cumulative incidence for study events during the follow-up period. To account for missing data, a multiple imputation analysis (MICE) was performed using the R statistical package using ten replicates and five iterations. All calculations were performed with R statistical software version 3.6.1.

### 2.5. Institutional Review Board Statement

This study was conducted according to the guidelines of the Declaration of Helsinki and approved by the Ethics Committee of the Primary Health Care University Research Institute Jordi Gol (protocol code P13/028). All patient records and information were anonymized and de-identified prior to analyses.

## 3. Results

### 3.1. Clinical Characteristics upon Inclusion

In our study, 146,506 patients with T2DM were screened with retinal photography, and their DR categories were recorded in their medical records. The cohort comprised 62,586 females (42.7%) and 83,920 males (57.3%). The mean duration of T2DM at baseline was 5.7 years, and the average of HbA1c was 7.2%. The median follow-up period was 6.96 years. Other clinical characteristics, such as BMI, GFR, lipid profiles, and other comorbidities, are included in Table 1. In the Appendix A shows the clinical characteristics of the subjects at baseline according to the subsequent development of different forms of DR.

### 3.2. Incidence of Diabetic Retinopathy and Risk Factors

During the follow-up period, 6881 cases of DR were diagnosed (corresponding to 4946 patients with mild NPDR, 1482 with moderate NPDR, 148 with severe NPDR, and 166 with PDR). Additionally, 139 patients developed DME. The overall cumulative incidence for the period was 4.7%, and the incidence rate was 6.99 per 1000 person-years. The incidence rates for the specific stages—mild NPDR, moderate NPDR, severe NPDR, PDR, and DME—were 4.98, 1.47, 0.15, 0.16, and 0.13 per 1000 person-years, respectively (Table 2).

The incidence of DR varied significantly across different risk factors. Patients with a diabetes duration greater than ten years had the highest incidence rate when compared with those with a duration of 0–5 years, with an HR of 2.62 (95% CI: 2.46; 2.78). Analysis of DR incidence between patients with strict glycemic control (HbA1c < 7%) and those with moderate control (7% ≤ HbA1c < 8%) revealed an HR of 1.55 (95% CI: 1.45; 1.65). A higher baseline HbA1c level was also associated with increased DR incidence. Other significant risk factors included smoking, hypertension, and CKD, all of which showed elevated incidence rates and HRs in the multivariate analysis (Table 3). Notably, obesity appeared to be a protective factor against DR, with an HR of 0.93 (95% CI: 0.89 and 0.98). The incidence of the different stages of DR (i.e., mild, moderate, and severe NPDR; PDR; and DME) according to risk factors is summarized in the Appendix A. Across these DR subgroups, our findings align with the overall risk trends for developing DR. Specifically, patients with a diabetes duration of more than ten years and elevated HbA1c levels consistently showed higher incidence rates across all DR stages.

The cumulative incidence for DR during the follow-up period is shown in Figure 1.

In the multivariate Cox regression analysis, the importance of the different risk factors was analyzed by adjusting several predictive models (Appendix A and Figure 2).

## 4. Discussion

Our study provides data on the incidence of DR from a large cohort of patients with T2DM within a primary care setting. Over an average follow-up period of almost 7 years, the overall incidence of DR was 6.99 per 1000 person-years, with cumulative incidence reaching 4.7%. These figures align with previous studies conducted on similar populations, reinforcing the growing concern regarding DR as a leading cause of visual impairment and preventable blindness in patients with diabetes [5]. 

The incidence rate in this study is consistent with that reported in research conducted on European populations. A systematic review by Li et al. (2020) reported an annual incidence of DR in Europe amounting to 4.6% (95% CI 2.3–8.8%) [26]. A study with a similar follow-up period reported a cumulative incidence of 16% at eight years with an annual incidence of 4.43% in a Spanish population [32]. A systematic review conducted by Sabanayagam et al. (2019) reported that the annual incidence of DR ranged from 2.2% to 22.3%. However, in the analysis of incidence by period of time, studies conducted after 2000 reported a similar incidence of DR (3.4–5.6%) [25]. Additionally, a recent study conducted in Spain by Romero-Aroca et al. (2022) found an incidence of 3.83% (2.01–6.89%) [23]. The differences may be attributed to the longer duration of the follow-ups and the inclusion criteria. 

The incidence of different DR stages observed in our work—ranging from mild NPDR to PDR—is consistent with previous data. However, some studies report incidence rates for different stages of DR that differ from those observed in our study. For instance, the incidence of PDR reported in the Nakuru Eye Study in Kenya [33] was 0%, while in the SN-DREAMS-II in urban India [34], it was 1.5%. These discrepancies may be attributed to the baseline characteristics of the participants, as some studies include patients who already exhibit some degree of DR at the start. Such differences highlight the variability in DR progression across different populations and the importance of considering initial disease status when comparing incidence rates [25].

The incidence of DME found in our study (0.13/1000 person-years) is significantly lower than the one reported in hospital-based studies. For example, Kristtinson et al. (1997) identified an incidence rate of 3.4% [35], while Klein et al. (1995) reported a higher incidence, 13.6%, over a 10-year period [36]. However, their findings were based on a cohort with pre-existing DR, which included individuals with type 1 diabetes and a significantly longer baseline duration of the condition. Additionally, most studies report prevalence rather than incidence; for example, Yau et al. (2012) [8] and Varma et al. (2014) [37] reported DME prevalence rates of 6.8% and 3.8%, respectively. These discrepancies reflect differences in diagnostic methodology, population characteristics, and healthcare settings, emphasizing the need for advanced imaging tools in primary care to improve DME detection.

Our findings show that the duration of diabetes and baseline HbA1c levels are critical risk factors for developing DR. Patients with a diabetes duration of more than five years had a significantly higher risk of developing DR (HR: 1.72; 95% CI: 1.63–1.82). Those with over ten years of disease progression faced an even greater incidence of DR (HR: 2.62; 95% CI: 2.46–2.78). These results are similar to findings from previous research [25]. This reinforces the well-established relationship between the duration of diabetes and the risk of developing DR, as longer exposure to hyperglycemia increases the risk of microvascular complications.

These results suggest that tight glycemic control and the management of other comorbid conditions may be crucial in delaying the onset and progression of DR in patients with T2DM. Likewise, patients with HbA1c levels above 10% faced a markedly increased risk of DR compared with patients with levels below 7%, with an HR of 4.23 (95% CI: 3.90–4.58). These findings are consistent with findings from other studies that demonstrate a linear relationship between poor glycemic control and the risk of DR [38,39].

In the multivariable analysis, after adjusting for several clinical variables, HbA1c control proved to be a critical factor in the incidence of DR in individuals with T2DM. Even when adjusting for factors such as diabetes duration, the elevated DR risk remained robust, underscoring the importance of maintaining optimal glycemic control to prevent disease progression. These results are consistent with the results of other studies, which also reported an increased risk of severe retinal complications in patients with poor glycemic control [24].

Our multivariate analysis also identified other risk factors, including smoking, hypertension, and CKD, which are known to exacerbate microvascular damage in patients with diabetes [40,41,42]. Interestingly, in our study, obesity appeared to serve as a protective factor against the development of DR, with an HR of 0.93 (95% CI: 0.89 and 0.98). This contrasts with the common perception of obesity being a universal risk factor and aligns with emerging research on the heterogeneity among T2DM subgroups. For instance, there is a specific subgroup, known as SIDD (Severe Insulin-Deficient Diabetes) that showed a higher risk of DR; it consists of patients with a severe insulin deficiency and lower levels of obesity, suggesting that this cohort may face a higher risk of developing complications like DR [43]. These insights suggest that DR risk factors in type 2 diabetes can vary significantly by patient subtype, pointing to the value of more personalized approaches in DR prevention.

The results of this study underscore the importance of maintaining tight glycemic control to delay the onset and progression of DR. The relatively low incidence of severe stages of DR in our cohort suggests that early detection due to regular screening and appropriate management can effectively reduce the risk of advanced DR stages. These findings support current clinical guidelines that advocate for routine retinal screenings for patients with T2DM and for optimum management of risk factors to prevent DR-related complications.

Our study has some strengths and limitations. One of the major strengths of this study is its large sample size and the extensive follow-up period. Notably, we dealt with patients with T2DM who attended primary care, most of whom followed the corresponding controls. The SIDIAP database has allowed for a comprehensive analysis of DR incidence, risk factors, and progression in a real-world primary care setting. Consequently, our results are highly relevant for clinical practice. Using a well-validated electronic medical record system [27,28] ensured the accuracy and completeness of the data collected.

However, this study also has some limitations. A major limitation of all studies based on database analyses is the quality and completeness of records. Inaccuracies or missing data could affect the validity of the findings. However, the SIDIAP database has repeatedly demonstrated reliability and robustness for this type of analysis. It has been specifically designed as a research tool for improving healthcare outcomes, and its use in numerous studies has shown that it provides high-quality data, particularly in the management of chronic disease, such as diabetes, making it a valuable tool for epidemiological research [28]. The retrospective nature of the data collection process may introduce biases related to the accuracy of medical records and the potential for the misclassification of DR severity. Additionally, while the cohort was large and representative of the population, the results may not be generalizable to other populations with different demographic or clinical characteristics. Another possible limitation of our study is that the detection of DME in primary care relies solely on retinal photography, which may lead to an underestimation of the true incidence of DME, since these patients are usually referred to ophthalmologist specialists, and these data were not available in our primary health care database when the analysis was conducted.

## 5. Conclusions

Over an average follow-up period of almost 7 years, the overall incidence of DR was 6.99 per 1000 person-years, with the cumulative incidence reaching 4.7%. Our study reinforces the critical importance of regular retinopathy screening for patients with T2DM, particularly after five years of disease and for those with poorly controlled HbA1c levels. The increased risk observed for patients with HbA1c levels above 7% and the substantially higher risk for those with levels exceeding 10% highlight the need for early and frequent monitoring to prevent the development of more severe stages of DR. Further studies are needed to assess the progression of diabetic retinopathy and the potential for vision loss over time.

## Figures and Tables

**Figure 1 jcm-13-07083-f001:**
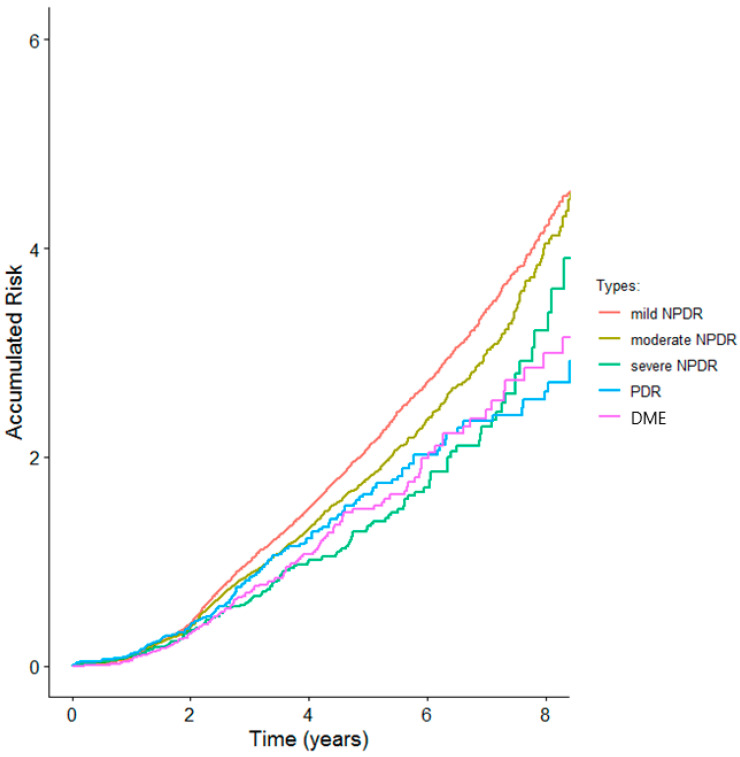
Cumulative incidence for DR during the follow-up period.

**Figure 2 jcm-13-07083-f002:**
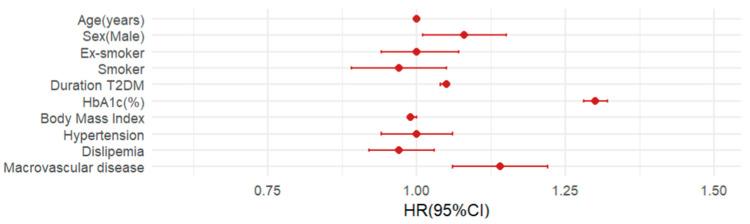
The fully adjusted multivariate model of the risk factors for diabetic retinopathy.

**Table 1 jcm-13-07083-t001:** Clinical characteristics of the subjects upon inclusion.

Variables	All [*N* = 146,506]
**Sociodemographic and toxic habits**	
Age (years), mean (SD)	65.4 (11.2)
Non-smoker	75,770 (52.9%)
Current smoker, *N* (%)	24,335 (17.0%)
Ex-smoker, *N* (%)	43,020 (30.1%)
**Clinical variables, mean (SD)**	
Diabetes duration, mean (SD)	5.7 (5.1)
BMI (kg/m^2^)	30.5 (5.1)
SBP (mmHg)	134 (15)
DBP (mmHg)	77 (10)
**Laboratory variables, mean (SD)**	
HbA1c (%)	7.2 (1.4)
Total cholesterol (mg/dL)	192 (40)
HDL cholesterol (mg/dL)	49 (13)
LDL cholesterol (mg/dL)	113 (34)
Triglycerides (mg/dL)	164 (117)
GFR (CKD-EPI; mL/min/1.73 m^2^)	58.2 (5.7)
UACR	30.0 (121)
**Comorbidities, N (%)**	
Dyslipidemia	91,293 (62.3%)
Hypertension	97,854 (66.8%)
Chronic kidney disease	30,234 (23.1%)
Coronary heart disease	13,319 (9.0%)
Stroke	8712 (5.9%)
Peripherical artery disease	5403 (3.6%)
Heart failure	5668 (3.8%)
**Concomitant treatment, N (%)**	
Antihypertensive drugs	107,409 (73.3%)
Antidiabetic drugs:	
No drugs	24,341 (16.6%)
NIADs combination	100,620 (68.7%)
INS mono therapy	2881 (1.9%)
NIADs + INS	18,664 (12.7%)

BMI: body mass index; SBP: systolic blood pressure; DBP: diastolic blood pressure; HbA1c: glycosylated hemoglobin; SD: standard deviation. CKD: chronic kidney disease; GFR: glomerular filtration rate; UACR: urine albumin/creatinine ratio; NIAD: non-insulin antidiabetic drugs; INS: insulin.

**Table 2 jcm-13-07083-t002:** Incidence of diabetic retinopathy during follow-up.

Group of Subjects	Patients (Number)	Person-Years	Follow-Up Time, Years (Median)	Events (Number)	Incidence Rate per 1000 Person-Years	Cumulative Incidence
Overall	146,506	984,941.3	6.96	6881	6.99	4.69
Mild NPDR	146,506	993,030	7.03	4946	4.98	3.38
Moderate NPDR	146,506	1,008,568	7.10	1482	1.47	1.01
Severe NPDR	146,506	1,014,329	7.12	148	0.15	0.10
PDR	146,506	1,014,170	7.12	166	0.16	0.11
DME	146,506	1,014,384	7.12	139	0.13	0.09

NPDR, non-proliferative diabetic retinopathy; PRD, proliferative diabetic retinopathy; DME, diabetic macular edema.

**Table 3 jcm-13-07083-t003:** Incidence of diabetic retinopathy based on risk factors.

Group of Subjects	Patients (Number)	Person-Years	Follow-Up (Mean)	Events (Number)	Incidence Rate per 1000 Person-Years	Cumulative Incidence	Hazard Ratio (95% CI)
**Female**	62,586	428,315.8	6.84	2821	6.59	4.51	Ref.
**Male**	83,920	556,625.5	6.63	4060	7.29	4.84	1.10 (1.04; 1.15)
**Smoker**							
Non-smoker	75,770	516,934.9	6.82	3526	6.82	4.65	Ref.
Ex-smoker	43,020	279,466.5	6.50	2063	7.38	4.80	1.06 (1.01; 1.12)
Current smoker	24,335	162,656.9	6.68	1140	7.01	4.68	1.02 (0.95; 1.09)
**Diabetes duration**							
0–5 years	79,526	539,958.9	6.79	25,608	4.83	3.28	Ref.
6–10 years	45,445	309,337.6	6.81	2536	8.20	5.58	1.72 (1.63; 1.82)
>10 years	21,535	135,644.6	6.30	1737	12.81	8.07	2.62 (2.46; 2.78)
**HbA1c (%)**							
<7%	70,255	461,234.5	6.57	2163	4.69	3.08	Ref.
7–8%	33,825	223,610.0	6.61	1611	7.20	4.76	1.55 (1.45; 1.65)
8.1–10%	18,316	120,436.4	6.58	1397	11.60	7.63	2.51 (2.35; 2.68)
>10%	6654	41,998.8	6.31	836	19.91	12.56	4.23 (3.90; 4.58)
**Obesity**							
BMI ≤ 30 kg/m^2^	64,253	419,357.66	6.53	30.14	7.19	4.69	Ref.
BMI > 30 kg/m^2^	60,669	402,438.0	6.63	2696	6.70	4.44	0.93 (0.89; 0.98)
**Comorbidities**							
Non hypertension	48,652	334,533.9	6.88	2306	6.89	4.74	Ref.
Hypertension	97,854	650,407.3	6.65	4575	7.03	4.68	1.01 (0.96; 1.06)
Non-macrovascular disease	122,397	834,773.6	6.82	5601	6.71	4.58	Ref.
Macrovascular disease	24,109	150,167.6	6.25	1280	8.52	5.31	1.24 (1.17; 1.32)
Non-chronic kidney disease	100,412	680,823.1	6.68	4370	6.51	4.35	Ref.
Chronic kidney disease	30,239	185,456.0	6.13	1659	8.95	5.49	1.35 (1.27; 1.42)

HbA1c, glycosylated hemoglobin; BMI, body mass index.

## Data Availability

Due to legal limitations, the dataset for this study is the property of DAP_Cat group; access to it can be requested by contacting the corresponding author Dr. Josep Franch-Nadal (email: josep.franch@gmail.com).

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
