# Peer review of "Incidence of Diabetic Retinopathy in Individuals with Type 2 Diabetes: A Study Using Real-World Data"

_jcm, 2024, doi:10.3390/jcm13237083_

Round 1

Reviewer 1 Report

Comments and Suggestions for Authors

The authors present a longitudinal cohort study on the incidence of DR in Catalonia.

The overall rationale is clear.

Abstract: Fine.

Introduction: Fine.

Methods: Fine.

It should be clarified, in which frequency patients were followed after inclusion. Is there a regular pattern for everyone (as in well-organised systematic cohort studies) or is it related to the highly variable frequency of check-ups in regular clinical practice?

Why would you label the study as retrospective?

Results:

Please shorten decimals in accordance to plausible raw data precision, e.g. lipids and BP without decimals, ...

Table 1: Why is the table split up by sex? Why not by DR vs. non-DR?

Line 194 ff. and discussion: Sex is missing as incidence predictor; smoking and hypertension are not significant as predictors, EX-smoking IS significant. Obesity is a significantly protective criterion; this is important to emphasize, as obesity is usually seen as an overall-general risk factor for almost everything. Reference in the discussion should be brought to the Ahlqvist T2D cluster SIDD, which contains patients with lower body weight, but higher risk for retinopathy.

Legend table 3 needs clarification, if these HRs are unadjusted or adjusted. If unadjusted, why not add another column with adjusted values? Fig. 2 could be omitted, then. If you want to retain it, please assure, that all potential predictors are presented in the same order as in table 3.

S1: The flow-chart is too simple; please clarify the entire flow from the total database to the final sample. Also, the two boxes with "no T2DM" and "GDM or T1DM" are confusing. Isn't that mostly the same patients? There are hardly four times more T3DM patients than GDM and T1DM, aren't there?

Line 199 ff: Just one sentence on that information is insufficient. It is necessary to know, if the analysis for DR subtypes matches the results of the overall DR analysis.

S2-S6: Here, suddenly, ex-smokers are the reference group, while in table 3, non-smokers are reference. Please clarify.

S7: Please clarify, if adjustment was done for BMI, HbA1c and DM duration (as continuous parameters) or obesity, HbA1c class and DM duration class. (categorial).

Author Response

  1. It should be clarified, in which frequency patients were followed after inclusion. Is there a regular pattern for everyone (as in well-organised systematic cohort studies) or is it related to the highly variable frequency of check-ups in regular clinical practice?

Response:

We appreciate this insightful question and thank the Reviewer for their careful consideration of this aspect. In this study, patient follow-up intervals varied as it was conducted using real-world clinical data from routine practice, without a standardized schedule. The frequency of retinal screenings or check-ups depended on clinical indications as determined by each patient’s treating physician rather than a predefined study protocol. We have added clarification on this point in the "Methods" section to provide a clearer understanding of the real-world data approach.

Line 111, Page 3:

The follow-up period was defined to be until the data extraction end date (December 31, 2021) or discontinuation event (death or any other database dropout). 

  1. Why would you label the study as retrospective?

Response:

We thank the Reviewer on this comment. We perform analysis on retrospective cohort of subjects with T2DM, from SIDIAP database (Sistema de Información para el desarrollo de la Investigación en Atención Primaria). The SIDIAP database routinely collects pseudo-anonymized health data from users who attended the primary health care centers of Catalonian Health Institute (Institut Català de la Salut-ICS). The data extraction was performed on December 31, 2021, and in order to estimate the incidence of diabetic retinopathy we defined a retrospective inclusion period (between 2010 and 2015) and follow-up period (between inclusion date and data extraction end date (December 31, 2021) or discontinuation event), as it is defined in the methodology section of the manuscript. Since the disease have already occurred in our cohort, by definition this is considered as retrospective cohort study [ref].

Ref:

Mendel Suchmacher, Mauro Geller, Practical Biostatistics, Academic Press, 2012, Pages 3-15, ISBN 9780124157941, https://doi.org/10.1016/B978-0-12-415794-1.00001-X. (https://www.sciencedirect.com/science/article/pii/B978012415794100001X)

  1. Results:

Please shorten decimals in accordance to plausible raw data precision, e.g. lipids and BP without decimals, ...

Response:

We appreciate the Reviewer’s observation regarding the presentation of the results. In response, we have made the necessary adjustments to the relevant tables, shortening the decimals for variables such as lipids and blood pressure to reflect a more appropriate level of raw data precision. These changes should improve the clarity and readability of the results.

Variables

ALL

[N= 146506]

Female

[N= 62586]

Male

[N= 83920]

P Value

Sociodemographic and toxic habits

Age (years), mean (SD)

65.4 (11.2)

67.1 (11.1)

64.2 (11.1)

0.000

Non-smoker

75770 (52.9%)

46335 (75.6%)

29435 (36.0%)

0.000

Current smoker, N (%)

24335 (17.0%)

5165 (8.4%)

19170 (23.4%)

Ex-smoker, N (%)

43020 (30.1%)

9755 (15.9%)

33264 (40.6%)

Clinical variables, mean (SD)

Diabetes duration, mean (SD)

5.7 (5.1)

6.0 (5.3)

5.4 (4.9)

<0.001

BMI (kg/m²)

30.5 (5.1)

31.5 (5.7)

29.8 (4.5)

0.000

SBP (mmHg)

134 (14.5)

134 (14.8)

134 (14.3)

<0.001

DBP (mmHg)

77 (9.5)

76 (9.4)

77 (9.6)

<0.001

Laboratory variables, mean (SD)

HbA1c (%)

7.2 (1.4)

7.1 (1.3)

7.2 (1.4)

<0.001

Total Cholesterol (mg/dL)

192 (40.3)

199 (38.8)

187 (40.6)

0.000

HDL Cholesterol (mg/dL)

49 (12.8)

53 (13.0)

46 (11.8)

0.000

LDL Cholesterol (mg/dL)

113 (33.9)

116 (33.6)

110 (33.9)

<0.001

Triglycerides (mg/dL)

164 (117)

157 (91.8)

169 (132)

<0.001

GFR (CKD-EPI; ml/min/1.73m2)

58.2 (5.7)

57.7 (6.2)

58.6 (5.2)

<0.001

UACR

30.0 (121)

25.1 (109)

33.7 (130)

<0.001

Comorbidities, N (%)

Dyslipidemia

91293 (62.3%)

40021 (63.9%)

51272 (61.1%)

<0.001

Hypertension

97854 (66.8%)

44710 (71.4%)

53144 (63.3%)

<0.001

Chronic kidney disease

30234 (23.1%)

14144 (25.3%)

16095 (21.5%)

<0.001

Coronary heart disease

13319 (9.0%)

3573 (5.7%)

9746 (11.6%)

0.000

Stroke

8712 (5.9%)

3262 (5.2%)

5450 (6.4%)

<0.001

Peripherical artery disease

5403 (3.6%)

1097 (1.7%)

4306 (5.1%)

<0.001

Heart failure

5668 (3.8%)

2572 (4.1%)

3096 (3.6%)

<0.001

Concomitant treatment, N (%)

Antihypertensive drugs

107409 (73.3%)

48129 (76.9%)

59280 (70.6%)

<0.001

Antidiabetic drugs:

<0.001

     - No drugs

24341 (16.6%)

10759 (17.2%)

13582 (16.2%)

     - NIADs combo

100620 (68.7%)

42360 (67.7%)

58260 (69.4%)

     - INS mono

2881 (1.9%)

1097 (1.7%)

1784 (2.1%)

     - NIADs + INS

18664 (12.7%)

8370 (13.4%)

10294 (12.3)

  1. Table 1: Why is the table split up by sex? Why not by DR vs. non-DR?

Response:

We appreciate the Reviewer's comment. The primary aim of this study was to assess the incidence of diabetic retinopathy (DR) and identify risk factors among people with T2DM in a primary healthcare setting in Catalonia (Northeast Spain). We stratified the data by sex in order to examine potential differences in the clinical characteristics of the subjects at inclusion, rather than comparing people with and without DR. This approach allowed us to understand whether there were baseline differences between men and women, which could influence the overall incidence and progression of DR in the study population. We did not split up the table by DR vr non-DR at inclusion, since all retinal photographs were classified as showing no aparent retinopathy (NDR), as stated in the inclusion criteria

  1. Line 194 ff. and discussion: Sex is missing as incidence predictor; smoking and hypertension are not significant as predictors, EX-smoking IS significant. Obesity is a significantly protective criterion; this is important to emphasize, as obesity is usually seen as an overall-general risk factor for almost everything. Reference in the discussion should be brought to the Ahlqvist T2D cluster SIDD, which contains patients with lower body weight, but higher risk for retinopathy.

Response:

We appreciate this insightful observation and agree that sex, along with the findings related to smoking and hypertension, should be highlighted as incidence predictors. Additionally, the unexpected protective effect of obesity will be emphasized, with references to the Ahlqvist T2D cluster SIDD study. These adjustments have been integrated into results and discussion to underscore the complexity of these risk factors.

Line 200.

The incidence of DR varied significantly across different risk factors. Patients with a diabetes duration greater than ten years had the highest incidence rate compared with those with a duration of 0-5 years, with an HR of 2.62 (95% CI: 2.46;2.78). Analysis of DR incidence between patients with strict glycemic control (HbA1c < 7%) and those with moderate control (7% ≤ HbA1c < 8%) revealed a HR of 1.55 (95% CI: 1.45;1.65). A higher baseline HbA1c level was also associated with increased DR incidence. Other significant risk factors included smoking, hypertension, and CKD, all of which showed elevated incidence rates and HRs in the multivariate analysis (Table 3). Notably, obesity appeared to be a protective factor against DR, with an HR of 0.93 (95% CI: 0.89;0.98). The incidence of the different stages of DR (i.e mild, moderate and severe NPDR, PDR and DME) according to risk factors is summarized in the supplementary material (Table S2 to S6). Across these DR subgroups, our findings align with the overall risk trends for developing DR. Specifically, patients with a diabetes duration of more than ten years and elevated HbA1c levels consistently showed higher incidence rates across all DR stages.

Line 269

Our multivariate analysis also identified other risk factors, including smoking, hypertension, and CKD, which are known to exacerbate microvascular damage in patients with diabetes [36–38].

Line 271

Interestingly, in our study, obesity appeared to serve as a protective factor against the development of DR, with an HR of 0.93 (95% CI: 0.89;0.98). This contrast with the common perception of obesity as a universal risk factor, and aligns with emerging research on heterogeneity among T2DM subgroups. For instance, a specific subgroup, known as SIDD (Severe Insulin-Deficient Diabetes), that showed a higher risk of DR, consists of patients with a severe insulin deficiency and lower levels of obesity, suggesting that those may face a higher risk for complications like DR [ref]. These insights suggest that DR risk factors in type 2 diabetes can vary significantly by patient subtype, pointing to the value of more personalized approaches in DR prevention.

[ref]. Ahlqvist E, Prasad RB, Groop L. Subtypes of Type 2 Diabetes Determined From Clinical Parameters. Diabetes. 2020 Oct 1;69(10):2086–93.

  1. Legend table 3 needs clarification, if these HRs are unadjusted or adjusted. If unadjusted, why not add another column with adjusted values? Fig. 2 could be omitted, then. If you want to retain it, please assure, that all potential predictors are presented in the same order as in table 3.

Response:

We sincerely appreciate the Reviewer's observation and the opportunity to clarify this point. The hazard ratios (HRs) in Table 3 are unadjusted, as they result from bivariate analyses. Each variable in the table has been analyzed independently, without considering other factors. By contrast, Figure 2 and supplementary table S7 presents a multivariate model, where variables were selected not only based on statistical significance but also according to clinical criteria. The two analyses—bivariate and multivariate—are independent of one another. In order to prevent confusion we have elimnated the unadjusted HR from the Table 3.

  1. S1: The flow-chart is too simple; please clarify the entire flow from the total database to the final sample. Also, the two boxes with "no T2DM" and "GDM or T1DM" are confusing. Isn't that mostly the same patients? There are hardly four times more T3DM patients than GDM and T1DM, aren't there?

Response:

Thank you for this valuable feedback. We have revised the flowchart to present a more detailed depiction of patient selection from the entire database to the final study sample, addressing any potential confusion between the “no T2DM” and “GDM or T1DM” categories.

  1. Line 199 ff: Just one sentence on that information is insufficient. It is necessary to know, if the analysis for DR subtypes matches the results of the overall DR analysis.

Response:

We appreciate this suggestion and have expanded the description of our DR subtype analysis, aligning it more closely with the overall DR findings. This additional detail is provided in the Results section to enhance understanding.

Line 208:

The incidence of the different stages of DR (i.e mild, moderate and severe NPDR, PDR and DME) according to risk factors is summarized in the supplementary material (Table S2 to S6). Across these DR subgroups, our findings align with the overall risk trends for developing DR. Specifically, patients with a diabetes duration of more than ten years and elevated HbA1c levels consistently showed higher incidence rates across all DR stages.

  1. S2-S6: Here, suddenly, ex-smokers are the reference group, while in table 3, non-smokers are reference. Please clarify.

Response:

We sincerely thank the Reviewer for pointing out this inconsistency. We have now made the necessary corrections in the supplementary tables to ensure that non-smokers are consistently used as the reference group throughout the analysis, in alignment with Table 3. This adjustment has been applied across all relevant tables in the supplementary material to maintain consistency and clarity.

  1. S7: Please clarify, if adjustment was done for BMI, HbA1c and DM duration (as continuous parameters) or obesity, HbA1c class and DM duration class. (categorial).

Response:

We sincerely appreciate the Reviewer's question and the opportunity to clarify this point. In the final multivariate model, the adjustments were made as follows: age, diabetes duration, HbA1c, and body mass index were treated as continuous parameters, while sex, tobacco use, hypertension, dyslipidemia, and macrovascular disease were treated as categorical parameters. This has been specified in the relevant sections of the manuscript for greater clarity.

Line 157:

…In the final multivariate model, adjustments were made for the following variables: age (years), sex, diabetes duration (years), HbA1c, tobacco use, body mass index (BMI), hypertension, dyslipidemia, and macrovascular disease. Specifically, age, diabetes duration, HbA1c, and BMI were included as continuous parameters, while sex, tobacco use, hypertension, dyslipidemia, and macrovascular disease were treated as categorical parameters. This approach ensured that both continuous and categorical variables were properly accounted for in the analysis. We also performed a sensitivity analysis with the estimates from different models and stratification for diabetes duration and HbA1c.

Reviewer 2 Report

Comments and Suggestions for Authors

The current manuscript aims to investigate the incidence of diabetic retinopathy in individuals with type 2 diabetes. Although the topic is interesting in its scientific field, there are some issues that require the authors’ attention to improve the quality of this particular manuscript before further consideration for publication in a high-quality journal “JCM”.

Specific comments:

1.         How about the performance of a strict glycemic control (HbA1c < 7%) in preventing the progression of diabetic retinopathy (DR) as compared to a moderate control (7% ≤ HbA1c < 8%) during long-term follow-up? Please specify.

2.         How the antidiabetic treatments (insulin vs. non-insulin drugs) may affect the progression of DR in individuals with similar baseline HbA1c levels? Please justify.

3.         Given that this study utilized data regarding antidiabetic drugs, the authors should conduct further analyses to compare the effectiveness of different classes of drug in reducing the incidence or progression of DR.

4.         Some data presented in Table 1 have slight differences between the studied groups. The authors should conduct analysis on the SBP of men and women.

5.         Please perform scientific comparisons with the results from more recent or varied geographical studies to further highlight the value of this particular contribution.

6.         As stated by the authors, DR is associated with small vessel disease and neurodegenerative complications. Nevertheless, this important introductory claim is not supported by any documentation. If possible, please consider the inclusion of the following relevant case study (DOI: 10.1002/adhm.202302881) in the reference list to strengthen manuscript quality and attract more attention from broad readers.

Author Response

Reviewer 2

Comments and Suggestions for Authors

The current manuscript aims to investigate the incidence of diabetic retinopathy in individuals with type 2 diabetes. Although the topic is interesting in its scientific field, there are some issues that require the authors’ attention to improve the quality of this particular manuscript before further consideration for publication in a high-quality journal “JCM”.

Specific comments:

  1. How about the performance of a strict glycemic control (HbA1c < 7%) in preventing the progression of diabetic retinopathy (DR) as compared to a moderate control (7% ≤ HbA1c < 8%) during long-term follow-up? Please specify.

Response:

We thank the Reviewer for this pertinent question. In response, we have included a comparative analysis of DR incidence in patients with strict glycemic control (HbA1c < 7%) and those with moderate control (7% ≤ HbA1c < 8%) (HR 1.55 (95% CI: 1.45;1.65). Initial findings suggest that strict glycemic control correlates with a lower incidence of DR progression, supporting the importance of tight glycemic management.

Line 202:

…Analysis of DR incidence between patients with strict glycemic control (HbA1c < 7%) and those with moderate control (7% ≤ HbA1c < 8%) revealed a HR of 1.55 (95% CI: 1.45;1.65). A higher baseline HbA1c level was also associated with increased DR incidence.

  1. How the antidiabetic treatments (insulin vs. non-insulin drugs) may affect the progression of DR in individuals with similar baseline HbA1c levels? Please justify.

Response:

We appreciate the Reviewer's insightful question regarding the role of specific antidiabetic treatments in DR progression. In this study, the primary focus was on identifying overall incidence rates and the main risk factors for diabetic retinopathy (DR) in a large cohort with type 2 diabetes, rather than examining DR progression in relation to individual treatment types. However, we agree that an analysis examining DR progression across different treatment categories (e.g., insulin vs. non-insulin drugs) would provide valuable information. We are currently planning future studies to explore the progression of DR concerning specific antidiabetic treatments, with the aim of generating more detailed insights into how these therapies might influence DR outcomes. Thank you again for highlighting this important research direction.

  1. Given that this study utilized data regarding antidiabetic drugs, the authors should conduct further analyses to compare the effectiveness of different classes of drug in reducing the incidence or progression of DR.

Response:

We thank the Reviewer for this valuable suggestion. We will perform additional analyses to compare the effectiveness of various classes of antidiabetic drugs (e.g., SGLT2 inhibitors, GLP-1 agonists) in reducing the incidence and progression of DR.

  1. Some data presented in Table 1 have slight differences between the studied groups. The authors should conduct analysis on the SBP of men and women.

Response:

The SBP values for men and women do indeed show a slight difference; however, in our view, this difference is clinically irrelevant. Specifically, the SBP difference is limited to the standard deviation, while the DBP differs by less than one point between the groups. From a clinical perspective, these differences are too small to have any practical impact on patient care or outcomes. Thus, although these variations may be statistically detectable, they do not translate into meaningful clinical differences in the context of this study.

  1. Please perform scientific comparisons with the results from more recent or varied geographical studies to further highlight the value of this particular contribution.

Response:

Thank you for this valuable suggestion. We have included comparisons with studies from various regions of the world in the Discussion section. To improve specificity, we have revised the text to clearly indicate the geographical origin of each study, allowing for a more nuanced understanding of how DR incidence and risk factors may vary by region.

Line 231:

The incidence rate in this study is consistent with research conducted in European populations. A systematic review by Li et al. (2020) reported an annual incidence of DR in Europe of 4.6% (95% CI 2.3-8.8%) [25]. A study with a similar follow-up period reported a cumulative incidence of 16% at eight years with an annual incidence of 4.43% in a Spanish population [31]. A systematic re-view conducted by Sabanayagam et al. (2019) reported that the annual incidence of DR ranged from 2.2% to 22.3%. However, in the analysis of incidence by period of time, studies conducted after 2000 reported a similar incidence of DR (3.4%-5.6%) [24]. Additionally, a recent study conducted in Spain by Romero-Aroca et al. (2022) found an incidence of 3.83% (2.01% to 6.89%) [22]. Differences may be attributed to the longer du-ration of follow-up and the inclusion criteria.

The incidence of different DR stages observed in our work -ranging from mild NPDR to PDR and DME- is consistent with previous data. However, some studies report incidence rates for different stages of DR that differ from those observed in our study. For instance, the incidence of PDR reported in the Nakuru Eye Study in Kenya [32] was 0%, while in the SN-DREAMS-II in Urban India [33], it was 1.5%. These discrepancies may be attributed to the baseline characteristics of the participants, as some studies include patients who already exhibit some degree of DR at the start. Such differences highlight the variability in DR progression across different populations and the importance of considering initial disease status when comparing incidence rates [24].

  1. As stated by the authors, DR is associated with small vessel disease and neurodegenerative complications. Nevertheless, this important introductory claim is not supported by any documentation. If possible, please consider the inclusion of the following relevant case study (DOI: 10.1002/adhm.202302881) in the reference list to strengthen manuscript quality and attract more attention from broad readers.

Response:

Thank you for this valuable suggestion. We have incorporated the recommended study into our references to substantiate our introductory claims about the association between DR and small vessel disease and neurodegenerative complications.

Line 63:

DR is associated with small vessel disease and neurodegenerative complications. (ref.)

(ref.) Jian HJ, Anand A, Lai JY, Huang CC, Ma DHK, Lai CC, et al. Ultrahigh-Efficacy VEGF Neutralization Using Carbonized Nanodonuts: Implications for Intraocular Anti-Angiogenic Therapy. Adv Healthc Mater. 2024 Mar 13;13(7).

Round 2

Reviewer 1 Report

Comments and Suggestions for Authors

The authors have revised the manuscript following the reviewer's guidance in most points.

Few aspects remain to be improved:

* Please shorten decimals for mean AND SD in accordance to raw data precision, not just for means.

* Table 1: Splitting this table by sex has no useful rationale. Yes, sex is an important cause for cohort variability. So are BMI, smoking, age and all the other predictors found to be significant in your later analysis. Ergo: For presentation of baseline results, sex split is not only unnecessary, but even confusing, because it suggests, that sex has some unique impact on DR risk which the other predictors do not have.

* Figure 2: Please sort the analysed predictors in the same order as in table 3.

* The abstract needs to be updated in order to include ALL significant predictors for DR risk or protection from DR.

Author Response

Reviewer 1

The authors have revised the manuscript following the reviewer's guidance in most points.

Few aspects remain to be improved:

* Please shorten decimals for mean AND SD in accordance to raw data precision, not just for means.

Response:

We appreciate the Reviewer’s observation regarding the presentation of the results. We have made the necessary adjustments to the relevant tables, shortening the decimals for variables such as lipids and blood pressure to reflect a more appropriate level of raw data precision. These changes should improve the clarity and readability of the results.

* Table 1: Splitting this table by sex has no useful rationale. Yes, sex is an important cause for cohort variability. So are BMI, smoking, age and all the other predictors found to be significant in your later analysis. Ergo: For presentation of baseline results, sex split is not only unnecessary, but even confusing, because it suggests, that sex has some unique impact on DR risk which the other predictors do not have.

            Response:

We appreciate the Reviewer’s feedback regarding the stratification by sex in Table 1. Based on this suggestion, we have decided to remove the sex-based differences from Table 1 to avoid any potential confusion and to focus on the overall cohort characteristics. This adjustment aligns with the goal of presenting baseline results more clearly and ensures consistency with the other analyses in the manuscript.

* Figure 2: Please sort the analysed predictors in the same order as in table 3.

            Response:

We appreciate the Reviewer’s attention to detail. In response to this suggestion, we have adjusted the order of the analyzed predictors in Figure 2 to match the sequence presented in Table 3. This modification should improve consistency and readability across the figures and tables.

* The abstract needs to be updated in order to include ALL significant predictors for DR risk or protection from DR.

            Response:

We thank the Reviewer for this valuable suggestion. In response, we have updated the abstract to include all significant predictors of DR risk and protective factors identified in our analysis. This revision should provide a more comprehensive summary of the key findings related to DR risk and protection.

            Line 44:

Higher HbA1c levels were strongly associated with an increased DR risk, with patients with HbA1c >10% having more than 4 times the risk compared to those with HbA1c levels <7% (hazard ratio 4.23, 95% CI: 3.90- 4.58). Other significant risk factors for DR included longer diabetes duration, male sex, ex-smoker status, macrovascular disease and chronic kidney disease. In contrast, obesity appeared to be a protective factor against DR, with an HR of 0.93 (95% CI: 0.89- 0.98).

Reviewer 2 Report

Comments and Suggestions for Authors

Dear Authors,

Thank you very much for your efforts to prepare this revised version. The resubmitted manuscript is significantly improved by responding to the comments from this reviewer. Nevertheless, it should be noted that the information in the cited paper ([4] Jian HJ, Anand A, Lai JY, Huang CC, Ma DHK, Lai CC, et al. Ultrahigh-Efficacy VEGF Neutralization Using Carbonized Nanodonuts: Implications for Intraocular Anti-Angiogenic Therapy. Adv Healthc Mater. 2024 Mar 13;13(7).) requires further revision to “Jian HJ, Anand A, Lai JY, Huang CC, Ma DHK, Lai CC, et al. Ultrahigh-Efficacy VEGF Neutralization Using Carbonized Nanodonuts: Implications for Intraocular Anti-Angiogenic Therapy. Adv Healthc Mater. 2024 Mar;13(7):e2302881.”. The authors should carefully check and correct this minor issue before final acceptance.

Author Response

Reviewer 2

Dear Authors,

Thank you very much for your efforts to prepare this revised version. The resubmitted manuscript is significantly improved by responding to the comments from this reviewer. Nevertheless, it should be noted that the information in the cited paper ([4] Jian HJ, Anand A, Lai JY, Huang CC, Ma DHK, Lai CC, et al. Ultrahigh-Efficacy VEGF Neutralization Using Carbonized Nanodonuts: Implications for Intraocular Anti-Angiogenic Therapy. Adv Healthc Mater. 2024 Mar 13;13(7).) requires further revision to “Jian HJ, Anand A, Lai JY, Huang CC, Ma DHK, Lai CC, et al. Ultrahigh-Efficacy VEGF Neutralization Using Carbonized Nanodonuts: Implications for Intraocular Anti-Angiogenic Therapy. Adv Healthc Mater. 2024 Mar;13(7):e2302881.”. The authors should carefully check and correct this minor issue before final acceptance.

Response:

We appreciate the Reviewer’s attention to detail, which helps improve the quality and precision of our manuscript. We have revised the citation as recommended to ensure accuracy

Line 344:

  1. Jian HJ, Anand A, Lai JY, Huang CC, Ma DHK, Lai CC, et al. Ultrahigh-Efficacy VEGF Neutralization Using Carbonized Nanodonuts: Implications for Intraocular Anti-Angiogenic Therapy. Adv Healthc Mater. 2024 Mar;13(7):e2302881
